# Mitochondrial Modulators: The Defender

**DOI:** 10.3390/biom13020226

**Published:** 2023-01-24

**Authors:** Emmanuel Makinde, Linlin Ma, George D. Mellick, Yunjiang Feng

**Affiliations:** Griffith Institute for Drug Discovery, Griffith University, Brisbane, QLD 4111, Australia

**Keywords:** mitochondria dysfunction, mitochondria health, mitochondria diseases, compounds

## Abstract

Mitochondria are widely considered the “power hub” of the cell because of their pivotal roles in energy metabolism and oxidative phosphorylation. However, beyond the production of ATP, which is the major source of chemical energy supply in eukaryotes, mitochondria are also central to calcium homeostasis, reactive oxygen species (ROS) balance, and cell apoptosis. The mitochondria also perform crucial multifaceted roles in biosynthetic pathways, serving as an important source of building blocks for the biosynthesis of fatty acid, cholesterol, amino acid, glucose, and heme. Since mitochondria play multiple vital roles in the cell, it is not surprising that disruption of mitochondrial function has been linked to a myriad of diseases, including neurodegenerative diseases, cancer, and metabolic disorders. In this review, we discuss the key physiological and pathological functions of mitochondria and present bioactive compounds with protective effects on the mitochondria and their mechanisms of action. We highlight promising compounds and existing difficulties limiting the therapeutic use of these compounds and potential solutions. We also provide insights and perspectives into future research windows on mitochondrial modulators.

## 1. Introduction

Mitochondria, widely understood to be the “power hub” of the cell, are organelles in eukaryotes responsible for most of the chemical energy supply required to fuel the cells’ complex web of biochemical reactions [1,2]. The mitochondria also perform crucial multifaceted roles in biosynthetic pathways, serving as an important source of building blocks for fatty acid, cholesterol, amino acid, glucose, and heme synthesis [3]. The mitochondria use fuels consumed by cells in the form of sugars, fatty acids, and amino acids to generate chemical energy [3,4]. For example, the mitochondria are an important hub for synthesizing amino acids such as glutamine, glutamate, alanine, proline, and aspartate. In addition, the initial step of gluconeogenesis, during which pyruvate carboxylase oxaloacetate is converted to malate, occurs in the mitochondria [3].

As shown in Figure 1, the mitochondria generate small molecule storage of chemical energy known as adenosine triphosphate (ATP) via electron transport-linked phosphorylation, otherwise known as oxidative phosphorylation (OXPHOS). The OXPHOS pathway utilizes five enzyme complexes in the inner membrane of the mitochondria to produce ATP as it progresses through the respiratory chain. These complexes include Complex I (NADH: ubiquinone oxidoreductase), Complex II (succinate dehydrogenase), Complex III (ubiquinol-cytochrome c oxidoreductase), Complex IV (cytochrome c oxidase), and Complex V (ATP synthase) [2,5,6,7]. In addition to its role at the core of energy metabolism, the mitochondrion is also an important site for calcium ion (Ca^2+^) storage and homeostasis while also playing a crucial role in cell apoptosis [2,8]. Furthermore, mitochondria are known to play a major role in the generation of reactive oxygen species (ROS), most of which are produced by Complex I and Complex III (on a smaller scale). Almost 90% of ROS generated in the mitochondria are essentially a by-product of the OXPHOS pathway [5,6,9]. The role of mitochondria in ROS generation is certainly noteworthy, especially with mounting research evidence establishing a link between disease progression in several neurodegenerative diseases and increased ROS production [10].

Oxidative stress resulting from the dysregulated generation of ROS can lead to impairments in the OXPHOS machinery, causing an imbalance in the mitochondrial redox potential and significant loss of mitochondrial functions (Figure 1). Impairments to mitochondrial function could result in decreased ATP generation, calcium overload, and unbalanced apoptosis [2,6,9]. Furthermore, disease conditions such as Huntington’s, Alzheimer’s, Parkinson’s, diabetes, several cancers, seizures, kidney failure, cardiomyopathy, and brain disorders, as well as normal aging processes, have been linked to failing mitochondrial functions [1,9,11,12,13,14].

As illustrated in Figure 2, the OXPHOS pathway starts with the entry of electrons into the respiratory chain via Complexes I and II, which are subsequently transferred to Complexes III and IV, respectively, and then used by Complex V to generate energy. [1,5,11,15].

Complex I, an L-shaped multimeric enzyme, catalyzes the first step of OXPHOS in the electron transport chain (Figure 2). Complex I binds and oxidizes NADH to generate two electrons which are used to reduce ubiquinone (coenzyme Q) to ubiquinol, an electron-rich form of coenzyme Q, which further transfers the electrons to Complex III [6,7]. Alongside complex I, complex II also binds and oxidizes FADH_2_ to generate two electrons, which are then transferred to ubiquinone [8,16]. Complex III transfers the electrons to cytochrome c (cyt c)and finally to Complex IV, which reduces O_2_ to H_2_O. The energy generated and released during this cascade of processes is then used to move protons from the mitochondrial matrix into the intermembrane space to generate an electrochemical potential utilized by Complex V to produce ATP from ADP and phosphate for cellular energy [6,7,11].

### 1.1. Mitochondria in Human Diseases

As previously enumerated, mitochondria play prominent roles in several processes at the cellular level. Consequently, the slightest alterations to any of these processes can ultimately lead to diseases and maladies in the human body. As a result, many vigorous attempts have been undertaken to explore the role of mitochondria in the pathogenesis of different diseases [2,9]. Despite the immense progress made to date, there are crucial things yet to be understood when it comes to the structure and function of the mitochondria as related to human diseases. Highlighted in the subsequent paragraphs is available evidence of mitochondrial dysfunction in selected human diseases.

#### 1.1.1. Mitochondria in Neurodegenerative Diseases and Ageing

Neurodegenerative diseases are a class of incurable diseases characterized by progressive degeneration and/or death of neurons, leading to the destruction of the nervous system. Neurons are well known to be high-energy demanding cells; hence it is not surprising that their health and functionality are closely linked to the mitochondria, which are the powerhouses of the cell [2,9,12]. Each neuron contains hundreds to thousands of mitochondria, and the central nervous system relies on normal mitochondrial functions for its high metabolic needs [17,18,19,20]. Although the etiology and pathogenesis of neurodegenerative diseases largely remain a mystery, research evidence has identified mitochondrial dysfunction as one of the key features of neurodegenerative disorders, such as Parkinson’s and Alzheimer’s diseases [17,21]. Therefore, a better understanding of mitochondrial function is critical to understanding and detangling the mysteries of neurological disorders.

In Parkinson’s disease (PD), extensive studies using cellular and animal models have implicated mitochondrial dysfunction, increased generation of ROS, and calcium imbalance as pivotal factors in the etiology of PD. In addition, a decline in mitochondrial Complex I activity has been reported in PD patient-sourced olfactory neurosphere-derived (hONS) cells, and involvement of PARK proteins in altered mitochondrial regulation has also been observed [12,15,22,23]. A thorough review discussing the role of mitochondria in PD was recently authored by Zambrano et al. [12].

In Alzheimer’s disease (AD), mitochondrial anomalies have also been identified as a common and consistent feature [21,24]. Accumulation of amyloid-β and phosphorylation of tau protein leading to the formation of neurofibrillary tangles are key hallmarks of AD, both of which have been linked to mitochondrial abnormalities [21,25]. Alterations in the morphology of the mitochondrion, enzyme, and DNA changes have also been observed in the brains of AD patients [21]. Similar to PD, oxidative stress induced by abnormal mitochondrial function is an early feature in AD [21,25]. In addition, partial inhibition of mitochondrial Complex I has been touted as a potential strategy for the treatment of AD [11].

Impairments in mitochondrial activity have also been reported in aging, which is one of the critical risk factors associated with most neurodegenerative diseases. Impairments such as mitochondrial DNA alteration, decreased proteasomal activity, increased ROS generation, and reduced activity of the OXPHOS machinery have all been linked to aging [10,17].

#### 1.1.2. Mitochondria in Metabolic Diseases

Lately, there has been an interesting surge in research findings linking mitochondrial impairments to metabolic diseases such as type II diabetes [25], insulin resistance [26], obesity, metabolic syndrome, stroke, non-alcoholic liver disease, and the list goes on [2,8]. A resounding finding in these reports is that mitochondrial dysfunction contributes significantly to oxidative stress and inflammation, which is a usual commonality in these metabolic diseases [9,27,28,29]. ROS homeostasis is pivotal to aerobic organisms as this ensures the balance between the rate and magnitude of production and subsequent elimination of ROS over time. Any imbalance of ROS homeostasis essentially overwhelms the mitochondrial electron transport chain and, by extension, OXPHOS, leading to a decline in mitochondrial contents and the rate of OXPHOS [9,27]. A decline in mitochondrial contents and the rate of OXPHOS, as well as the modification of mitochondrial dynamics in key organs associated with metabolic diseases, have been implicated in the etiology of several metabolic diseases [9,26].

#### 1.1.3. Mitochondria in Cancer

Although the role of mitochondria in cancer and tumor development is yet to be fully understood, mitochondrial defects have long been implicated in the etiology of cancers and tumors [30]. One important hallmark of cancer is the Warburg effect, which involves the reprogramming of ATP generation via the OXPHOS pathway to aerobic glycolysis [30,31]. In normal cells, the common mechanism for ATP generation is glucose metabolism via the OXPHOS pathway in mitochondria. However, even in the presence of functional mitochondria, most cancer cells bypass the mitochondria and rewire their metabolism to produce needed energy through aerobic glycolysis, which is less efficient and involves a high rate of glucose uptake and glycolysis followed by lactate formation [32,33]. This ‘selfish’ reprogramming enhances the progression and proliferation of cancer and tumor cells through the overexpression of glucose transporters, speedy inefficient production of ATP to meet energy demands, and accumulation of lactate which aids tumor progression and acidosis [33,34].

Since the 1920s, when the Warburg effect was first documented, several studies have reported defective mitochondrial respiration and mutated or low copies of mitochondrial DNAs in various cancers, including adenocarcinoma, breast, colon, prostate, head, and neck cancers [35,36,37]. Another widely reported hallmark of cancer is the excessive generation of ROS, which are commonly a by-product of the mitochondria-mediated metabolic process [30,37,38,39]. The mitochondria have also been established as a proven target for cancer treatment with a handful of FDA-approved mitochondrial-targeted compounds and several others at different stages of preclinical and clinical trials. Notable examples include metformin, mitoxantrone, cisplatin, and ME344 [30].

#### 1.1.4. Mitochondria and infectious diseases

Beyond their conventional role as the cell’s energy hub, mitochondria also play a crucial role as a signaling platform for innate immunity against infectious microbes, and the role of mitochondria in infectious disease has been extensively documented [40,41,42,43]. Major host responses against infections depend on mitochondrial functions, and receptors of the innate immune system can detect compromises in mitochondrial functions, subsequently triggering an immune response [41,42]. In addition, pathogens exploit mitochondrial functions to influence their survival and evade immunity by affecting OXPHOS and mitochondrial dynamics and disrupting communication between the mitochondria and nucleus [40,41,43].

During infection, pathogens are detected by pattern-recognition receptors (PRRs), which can recognize pathogen-associated molecular patterns (PAMPs) such as flagellins, liposaccharides, proteins, mannose, and nucleic acids, as well as danger-associated molecular motifs (DAMPs) such as cardiolipin, ROS, mitochondrial DNA and n-formyl peptide [40,42]. Mitochondrial DAMPs are released into the cytosol as a result of infections, injuries, or loss of mitochondrial homeostasis [40,42].

DAMPs and PAMPs can be detected by PRRs to trigger innate immune responses against viruses, bacteria, and other infectious pathogens [42,44]. PRRs are classed into four families, which include toll-like receptors TLRs, (NOD)-like receptors (NLRs), C-type lectin receptors (CLRs), and retinoic acid-inducible gene I (RIG-I)-like receptors (RLRs) [42]. In viral infections such as influenza, PAMPs are recognized by RLRs, which interact with mitochondria antiviral signaling protein (MAVs) in the mitochondrial membrane to trigger the production of pro-inflammatory cytokines and type 1 interferon as an immune response [40,42,43]. During bacterial infection, TLRs are stimulated by bacterial PAMPs to induce the release of mitochondrial ROS to initiate antibacterial defense, resulting in the killing of pathogenic bacteria [42,45].

### 1.2. Materials and methods

The aim of this review is to provide a comprehensive insight into compounds with therapeutic potential on mitochondrial functions and their mechanisms of action, with a focus on compounds that can modulate the mitochondria such that mitochondrial dysfunction is mitigated or prevented altogether. To achieve this, an extensive literature search was conducted on PubMed, Science-Direct, and Google-Scholar databases using the following search terms:

“Mitochondria,” “Mitochondrial Complex,” “Mitochondria Health Disease,” “Mitochondrial Dysfunction,” “Mitochondrial Dysfunction Compounds.”

As a result of our search, we found 61 compounds (Table 1) with protective effects on the mitochondria. Further searches were conducted exclusively on PubMed using the name of each of the 61 compounds and mitochondria as keywords. This was performed to discover multiple mitochondrial modulating activities of any compound that might not have been covered in the first round of searches and to explore detailed mechanistic studies of each compound.

All 61 compounds were gathered from articles published in the last 20 years, and all consulted articles were thoroughly read to extract relevant information, such as the disease model used for the bioactivity study, the dose administered, mitochondria-related activities, and mechanisms of action. In Table 1, we present all 61 compounds, the disease model, effective doses, mitochondria-related targets, mode of action, and references for each compound.

## 2. Mitochondrial Modulators, Mechanisms, and Targets

In Table 1, we present 61 mitochondrial modulators which are able to protect the mitochondria from toxic insults and/or improve mitochondrial function. This collection of compounds includes 52 natural products (Figure 3, Figure 4, Figure 5, Figure 6 and Figure 7) and nine synthetic compounds (Figure 8). It is noteworthy that 31 of the 52 natural products discussed are phenolic compounds, which represent 50.8% of the total (Figure 3 and Figure 4). This is unsurprising given that phenolic compounds are renowned for their excellent antioxidant activity and the fact that oxidative stress is one of the major indicators of mitochondrial dysfunction [166,167]. Others include four alkaloids (Figure 5), eight terpenes (Figure 6), one organic acid, one amine, one cyclic polyketide, one lactone, one benzochromone, and one coumarin derivative (Figure 7).

The biological activities of the compounds summarized in Table 1 were evaluated in cellular or animal models or a combination of both. Our search revealed that 83.6% were tested using at least one cell line, with SH-SY5Y cells accounting for 41% of the studies. This is unsurprising because the human neuroblastoma SH-SY5Y cell line is a common in vitro model for PD and other neurodegenerative diseases associated with mitochondrial dysfunctions [168,169,170].

Generally, compounds showed multiple modes of action, exerting their protective effects on the mitochondria by (1) restoring oxidative balance by inhibiting the production of ROS or blocking the harmful effects of ROS and increasing the activity of antioxidant enzymes, (2) modulating apoptotic markers, (3) promoting ATP synthesis, (4) enhancing the activities of mitochondrial complexes, mitochondrial biogenesis and restoration of normal mitochondrial morphology in the presence of mitochondria toxins such as rotenone, 6-OHDA and MPP^+^ [46,76,82,114,171]. The numerical distribution of the compounds based on structural class and mechanism is shown in Figure 9. Inhibition of ROS is a common feature in all 61 compounds; while 45 of the compounds have anti-apoptotic activity, 33 improved ATP synthesis, 24 increased the activities of complexes, 22 promoted mitochondrial biogenesis, and eight restored normal mitochondrial morphology.

### 2.1. Antioxidative Mechanisms of Mitochondrial Modulators

Given that 90% of ROS is generated by mitochondria, oxidative stress is one of the major hallmarks of mitochondrial dysfunction [2,172]. All compounds listed in Table 1 displayed antioxidant activity through multiple mechanisms. In combination with the reduction in ROS generation and lipid peroxidation (LPO), the effect of a compound on the activity of enzymes that regulate free radical scavenging and oxidative balance is a major way to determine its antioxidant activity [129,173]. In normal cellular conditions, super-oxide dismutase (SOD) converts superoxide radical to H_2_O_2_, and enzymes such as catalase (CAT) and glutathione peroxidase (GPX) reduce mitochondrial H_2_O_2_ by converting it to H_2_O. Hence the ability of any compound to increase the activity of these enzymes is taken as an indicator that the compound possesses antioxidant potential [114,174].

Nuclear E2-related factor 2 (Nrf2) is one of the most pivotal cell defense mechanisms against stressors. It is a transcription factor that signals the expression of oxidative enzymes and stimulates the increased expression of antioxidant genes in response to oxidative stress. Consequently, any disruption to the activation of the Nrf2-mediated antioxidant response exposes the cells and renders the mitochondria more sensitive to deleterious pro-oxidants and electrophiles [175]. As a result of its crucial cytoprotective role against the effects of oxidative stress, Nrf2 is now a well-known drug target in many neurodegenerative diseases, most of which are associated with mitochondrial dysfunction [23,175,176]. Hence, the ability of a compound of interest to activate Nrf2-mediated antioxidative response is relevant for assessing its antioxidant capacity as a mitochondrial modulator [114,171]. Compounds **1**, **2**, **7–9**, **14**, **16**, **18**, **21**, **23**, **24**, **27**, **29**, **32**, **37**, **42**, **46**, **53**, and **54** activate the Nrf2-mediated antioxidative response.

Excessive ROS generation is known to cause disruptions to the OXPHOS pathway and electron transport chain, leading to defects in mitochondrial respiration, ATP production, depletion of mitochondrial complexes, and collapse of mitochondrial membrane potential (ΔΨm) [23,175,177]. Nrf2, when activated, amplifies ATP production and ΔΨm by boosting substrate availability for OXPHOS, leading to enhanced activity of the mitochondrial complexes [175]. The ability of a metabolite to induce blockage of the mitochondrial permeability transition pore (mPTP) is also a very instructive parameter when measuring the extent of mitochondrial oxidative stress. This can be achieved by inhibiting cyclophilin D (CYPD), a key enzyme that regulates the opening and closing of mPTP [114,178]. The opening of mPTP leads to mitochondrial swelling and has been implicated as one of the causes of the loss of mitochondrial function in neurodegeneration. The opening of mPTP occurs as a result of the collapse of the ΔΨm, leading to the continuous burst of ROS into mitochondria, causing oxidative stress [88,178,179,180] and mitochondrial permeability transition (MPT) driven necrosis. MPT-driven necrosis is a type of regulated cell death characterized by uncontrolled loss of post-mitotic cells and can be delayed by inhibition of CYPD [181,182]. All the compounds listed in Table 1, except boswellic acid (**37**) and MHY-1684 (**56**), were reported to restore ΔΨm while sarain A (**32**), ellagic acid (**23**), salvianolic acid A (**30**), kaempferol (**3**), lycopene (**41**), FLZ (**55**) induced the blockage of the mPTP. Although not listed in Table 1 because it is out of the 20-year coverage of this review, it is pivotal to mention that cyclosporin A; an FDA-approved immunosuppressant medication is a proven and well-known MPT inhibitor and a promising potential mitochondrial-targeted neuroprotective agent [183,184].

### 2.2. Inhibition of Apoptosis

Inhibition of apoptotic pathways through the modulation of apoptotic markers is another mechanism commonly reported in compounds discussed in this review. This is unsurprising because oxidative stress is a known precursor to apoptosis, so mitochondria play a key role in maintaining cellular apoptotic balance [2]. Typically in cells, apoptosis is triggered by activating pro-apoptotic proteins belonging to the Bcl-2 family, e.g., Bax and Bak, which translocate to mitochondria to induce the release of cytochrome c into the cytosol [185]. The release of cytochrome c promotes the activation of caspase-3, -6, -7, and -9, which subsequently initiates cell death [115,185]. However, the Bcl-2 family includes several anti-apoptotic members such as Bcl-2, Bcl-xL, Mcl-1, and Bcl-w. For the maintenance of proper cellular function, it is important to establish a steady balance between pro-apoptotic and anti-apoptotic markers [115,185]. Compounds **2–5**, **8–9**, **12–15**, **17–21**, **23–26**, **28–30**, **34–35**, **37–41**, **43–45**, **48**, **51–53**, **58**, **59**, and **61** all showed anti-apoptotic activity by modulating these markers.

The mitochondria are pivotal to keeping apoptotic balance, and disruptions to this balance have been implicated in diseases such as PD and AD [115,148,185]. The PI3K/Akt/GSK-3β pathway also promotes the expression of anti-apoptotic and pro-apoptotic proteins. The kinase PI3K releases phosphatidylinositol-3,4,5-trisphosphate (PIP3), which activates Akt by promoting the translocation of Akt to the plasma membrane [115,148]. Activation of Akt then induces the expression of anti-apoptotic proteins such as Bcl-2, while GSK-3β, on the other hand, is a prerequisite for the activation of p53, which induces the expression of pro-apoptotic proteins such as Bax [185,186,187]. PI3K/Akt also inhibits the expression of serine-threonine kinase GSK-3β, a critical effector of PI3K/Akt cellular signaling and activator of neuronal apoptosis. Activation of the PI3K/Akt pathway ameliorates apoptosis through the phosphorylation of GSK-3β by AKT [185,186,187]. Compounds **7**, **19**, **27**, **30**, **31**, **35**, **40**, and **56** demonstrated anti-apoptotic activity by targeting the PI3K/Akt/GSK-3β pathway.

### 2.3. Mitochondrial Biogenesis and Mitophagy

Mitochondrial homeostasis is maintained by keeping the mitochondrial pool in a cell at a steady state through the simultaneous propagation of new mitochondria (mitochondrial biogenesis or mitogenesis) and the removal of old and damaged mitochondria through a process called mitophagy [188,189]. Mitogenesis is a complex process with at least 150 proteins involved [190,191]. However, there are three key interdependent markers known to play a crucial role in this process: peroxisome proliferator-activated receptor-γ coactivator-1α (PGC1α), silent mating-type information regulation proteins (Sirt), and AMP-activated kinase (AMPK) [110,190,191]. PGC1α is a transcriptional factor that binds to Sirt3 promoters, interacting with Nrf2 to facilitate the upregulation of antioxidants [190]. Sirts can modulate the induction of several markers, such as PGC1α, to enhance the expression of antioxidant enzymes such as SOD [190,192,193]. Compounds **1**, **3**, **5**, **10**, **14**, **18**, **21–23**, **27**, **28**, **30**, **31**, **36**, **41**, **43–46**, **48**, and **54** were reported to modulate the AMPK/ PGC1α/Sirt pathway.

AMPK is inhibited by ATP, and the AMP/ATP ratio gives a sensitive indication of the metabolic state of a cell [194]. AMPK, when activated by mitochondrial ROS, promotes mitophagy, which may lead to a reduction in mitochondrial number, a decrease in ATP levels, and a subsequent increase in AMP/ATP ratio [194,195,196]. The PTEN-induced kinase 1 (PINK1) pathway-Parkin pathway also plays a crucial role in the mitophagy of weak mitochondria and maintenance of mitochondrial homeostasis [196,197,198,199]. Several studies have linked the accumulation of weak mitochondria in cells due to decreased mitophagy to deficient levels of PINK1 and Parkin [196,197]. Compounds quercetin (**1**), resveratrol (**14**), and ligustilide (**49**) were reported to modulate PINK1/Parkin.

### 2.4. Other Effects of Mitochondrial Modulators

In addition to the biological activities earlier discussed, 33 compounds (**1**, **3–5**, **7**, **10**, **15**, **17**, **22–24**, **26–32**, **39**, **41**, **43**, **46–48**, **51**, **58**, **59**) were reported to increase ATP synthesis, 24 (**4–7**, **10**, **11**, **13**, **14**, **16**, **17**, **23**, **24**, **28**, **30**, **34-37**, **41**, **45**, **46**, **51**, **55**, **57**) enhance the activity of mitochondria complexes, and 8 (**1**, **10**, **11**, **27**, **29**, **31**, **43**, **50**) restore mitochondrial morphology. In addition to their reported antioxidant activity and increase in MMP, 7,8-dihydroxyflavone (**6**) and diphenyl diselenide (**57**) increased the level of mitochondrial complexes, while β-lapachone (**47**) improved ATP synthesis [141,154]. However, sarain 2 (**33**) did not show any other activity apart from inhibiting ROS and increasing MMP [114].

Prohibitins (PHB) are evolutionarily conserved proteins ubiquitously expressed in eukaryotic cells and localized in the nucleus, cytosol, and mitochondria [200,201]. Large assemblies of homologous prohibitin members, prohibitin 1 (PHB1) and prohibitin 2 (PHB2), have been identified in the inner mitochondria membrane [201,202]. These mitochondrial PHB subunits play an important role in cell proliferation, mitochondrial biogenesis, mitochondrial dynamics, cell apoptosis, and senescence [200,201,203]. PHB impairments have been reported in aging, cancer, neurodegenerative, kidney, cardiovascular, and metabolic diseases, in which significant loss of mitochondrial function has been proven [200,201,202,203]. The marine natural product, aurilide which binds to PHB1, and a synthetic molecule, fluorizoline, which binds to PHB1 and PHB 2 are known potent PHB-binding compounds which have been reported to induce apoptosis and mitochondrial fragmentation [204,204,205]. Fluorizoline and aurilide have not been included in Table 1 since this review is focused on compounds that are able to mitigate or prevent mitochondrial dysfunction.

### 2.5. The Standouts

The most outstanding compound of all 61 discussed is perhaps rasagiline (**58**), a well-known monoamine oxidase inhibitor. Rasagiline is a novel propargylamine and an approved treatment for PD either as a monotherapy or in combination with other treatments such as levodopa [206,207]. Rasagiline was reported to attenuate mitochondrial dysfunction by reducing oxidative stress, preventing MMP collapse, suppressing apoptosis by lowering the release of cytochrome C, and improving ATP synthesis [158]. This is quite interesting because there are well-established links between PD and mitochondrial dysfunction [208], suggesting that rasagiline has multifaceted modes of action as far as the treatment of PD is concerned. The compound was also highly potent, showing excellent activity at a concentration range of 10 μM to 10 nM [158]. Rasagiline prevented ΔΨm collapse [158] and caused a three-fold increase in ATP level at 10 μM and 1 μM as well as a two-fold increase at 100 nM and 10 nM [159]. The compound also completely suppressed cell death at 10 μM and 1 μM by reducing cytochrome c [158,159]. Rasagiline is also currently trialed for treatment in AD, with results from the proof of concept of the phase II trials recently published [209].

Melatonin (**48**), a nocturnal hormone in the brain produced by the pineal gland, is also approved as Circadin^®^, a prolonged-release melatonin tablet for the treatment of insomnia, a common comorbidity of neurological disorders [210,211]. Additionally, a randomized phase II clinical trial of melatonin in the treatment of AD was successfully completed in 2013, with patients treated with prolonged-release melatonin showing significantly improved cognitive performance [212]. Melatonin also displayed a significantly better potency compared to the other compounds, second only to rasagiline at the tested concentration of 500 nM [142]. At 500 nM, melatonin significantly reduced ROS and caspase-3 in porcine oocytes treated with rotenone to levels comparable to that of the untreated group and enhanced mitochondrial biogenesis by upregulating PGC1α/SIRT while also increasing MMP and ATP synthesis [142].

The flavonoids, quercetin (**1**), and resveratrol (**14**) display the best coverage for biological activities discussed. They both have antioxidant potentials, activate the Nrf2 pathway, improve MMP, reduce apoptosis, and improve mitochondrial biogenesis and ATP synthesis [46,49,76,78]. In addition, Genistein (**8**) and asiatic acid (**38**) are by far the most active of the 52 natural products at the cellular level. Genistein reduced oxidative stress and improved MMP in a dose-dependent fashion in the concentration range from 10 pM to 100 nM in cells [66]. Asiatic acid, at 10 nM, caused a 75% decline in ROS generation, a 30–50% decline in pro-apoptotic markers such as Bax, cytochrome c, caspase-3, -6, -8, -9, and a 40% increase in Bcl-2 level when compared to cells treated with rotenone [125].

## 3. Conclusions and Future Directions

In this review, we have highlighted the multifaceted roles of the mitochondria as well as their relevance in health and disease. We enumerated 61 compounds, 52 of which are natural products, while nine are synthetic compounds. The fact that most of the compounds discussed are natural products underscores the well-known concept that nature holds vastly untapped therapeutic potential and will continue to play a critical role in drug discovery [213]. It cannot be overemphasized that tremendous advances have been made in natural product drug discovery. However, attention to natural products has declined in the past two decades due to challenging isolation and screening techniques, especially in high-throughput assays against molecular targets [213,214]. Despite these challenges, the field remains exceptionally viable, with millions of species unexplored and more metabolites waiting to be discovered [215]. Additionally, there is also the potential to improve current practices and approaches to natural products-based screening through innovative technological advances that are less arduous, time-saving, and with larger yields such as metabolomics, genome mining driven isolation, enhanced microbial culturing and biosynthetic engineering strategies [213].

Although unsurprising, it is noteworthy that 29 of the 61 compounds discussed in this article are phenolic compounds, many of which are considered safe and already available as dietary supplements [216,217]. Notable examples are resveratrol, quercetin, catechin, and kaempferol, but none of these have become approved commercially available drugs [217]. One major challenge with phenolic compounds is their low bioavailability; hence it is difficult to reproduce the in vitro biological activities of phenolic compounds in vivo [216,217,218]. Catechin, for instance, has very low bioavailability when taken orally, with its plasma concentration reported to be about 50 times lower than the concentration required to reproduce levels of bioactivity reported in vitro [218,219]. Studies have shown that absorption of quercetin in humans after ingestion can be as low as 2% [218], curcumin 5% [220,221], and resveratrol 70% but with bioavailability at trace levels (less than 1%) [222,223]. Consequently, there is a need for more in-depth studies into the pharmacokinetics of these compounds with the aim of enhancing their absorption and bioavailability through the development of more effective drug delivery systems such as micelles, liposomes, and nanoparticles. There is also the potential of developing analogous compounds through modification of existing phenolic compounds such that their absorption and bioavailability are improved without loss of bioactivity. Genistein is potentially a good candidate for absorption and bioavailability since it has high efficacy and potency in vitro at concentrations below nanomolar levels to the tune of 10 pM [66]. This may imply that a low concentration of genistein is needed to reproduce its activity in vivo without the problem of low bioavailability and absorption. However, this notion remains a theoretical suggestion and is subject to further investigations.

All compounds discussed in this review showed antioxidant activities. Hence, it is evident that alleviation of oxidative stress is the primary mechanism and mode of action of mitochondrial modulators, followed by inhibition of apoptosis reported in 45 of the compounds. Only 33 compounds improved ATP synthesis, while 22 improved mitochondrial biogenesis, and 24 enhanced the activity of mitochondrial complexes, suggesting that there is still room further to establish the mitochondria-modulating activities of the untested compounds.

Another issue worth considering in the search for mitochondrial modulators is the complexity of the mitochondria and the fact that mitochondria are tissue-specific organelles [224]. As a consequence of the specificity and heterogeneity of the mitochondria in different tissues, the mitochondria may show different morphology, distinct biochemical properties, and varied interactions with other intracellular organelles [224,225]. Notably, live imaging techniques have been used to show alterations in mitochondrial ROS, calcium homeostasis, membrane potential, and redox state in mitochondria isolated from different cells or tissues [226]. Furthermore, mitochondria in various tissues may also display different responses and sensitivity to molecules [225]. Hence, it might be interesting to consider if these molecules retain their biological activities across various cell lines and tissues and how that might affect the utility of the compounds in treating diseases associated with mitochondria dysfunction.

## Figures and Tables

**Figure 1 biomolecules-13-00226-f001:**
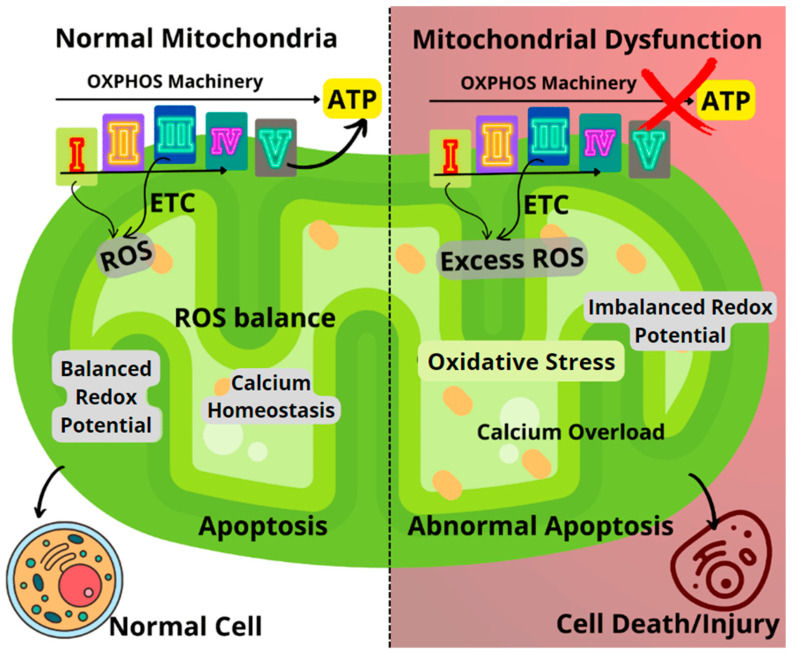
Comparison between healthy and dysfunctional mitochondria, highlighting the key mechanisms of mitochondrial dysfunction. ETC: Electron transport chain; ROS: Reactive oxygen species.

**Figure 2 biomolecules-13-00226-f002:**
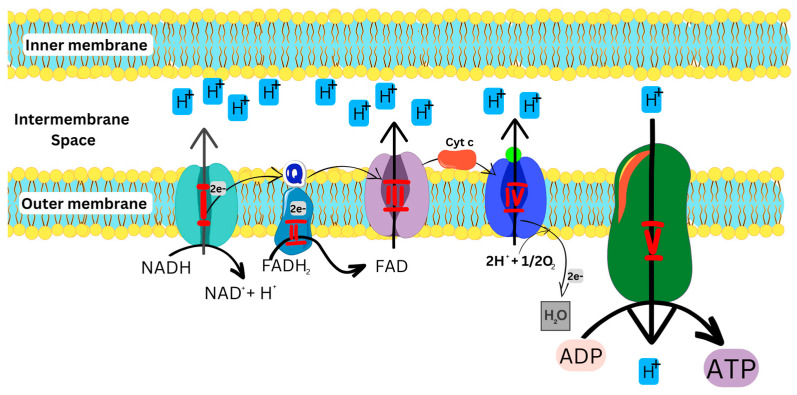
OXPHOS pathway, showing the transfer of electrons in the ETC to produce ATP. NADH: Reduced nicotinamide adenine dinucleotide; FADH: Reduced flavin adenine dinucleotide; ADP: Adenosine diphosphate; ATP: Adenosine triphosphate.

**Figure 3 biomolecules-13-00226-f003:**
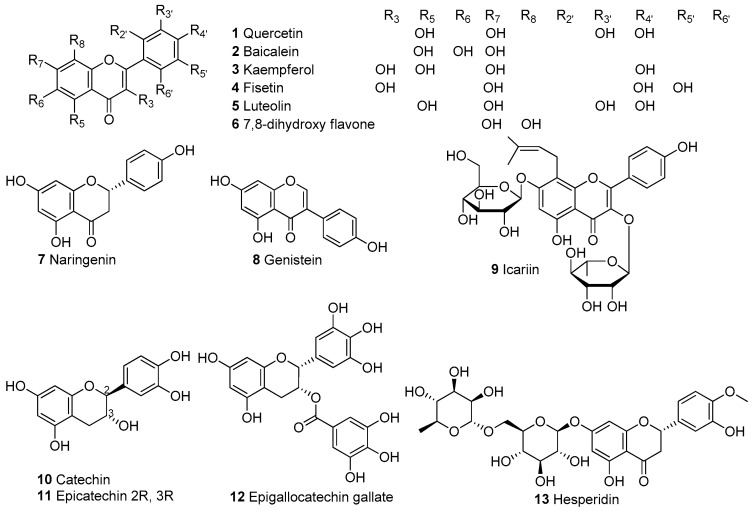
Chemical structures of flavonoids.

**Figure 4 biomolecules-13-00226-f004:**
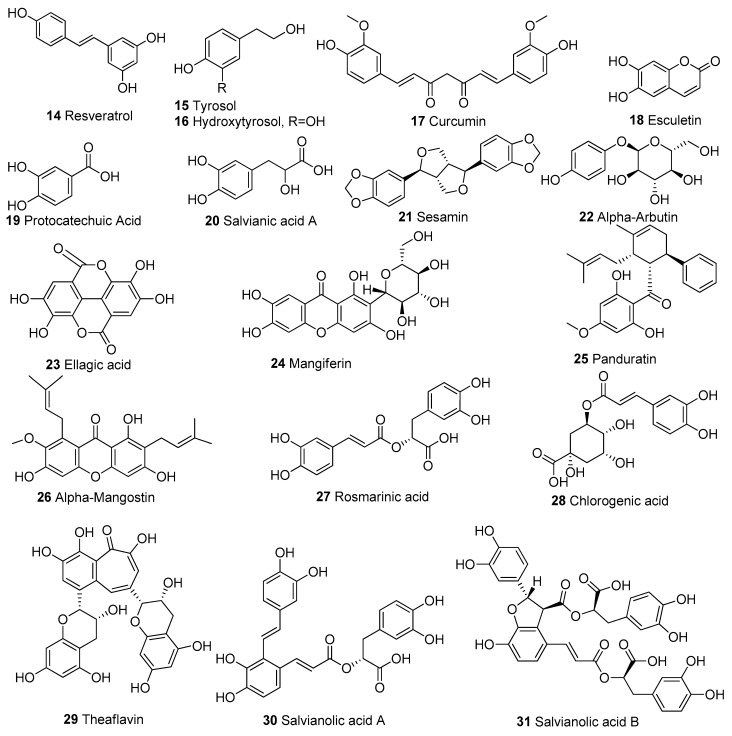
Chemical structures of other phenolic compounds apart from flavonoids.

**Figure 5 biomolecules-13-00226-f005:**
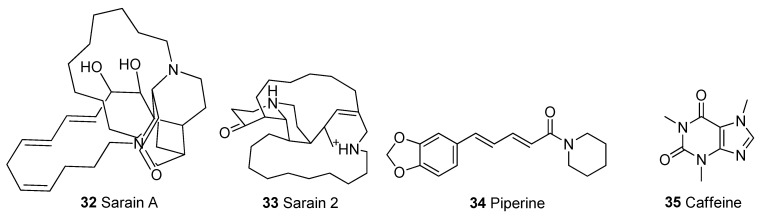
Chemical structures of alkaloids.

**Figure 6 biomolecules-13-00226-f006:**
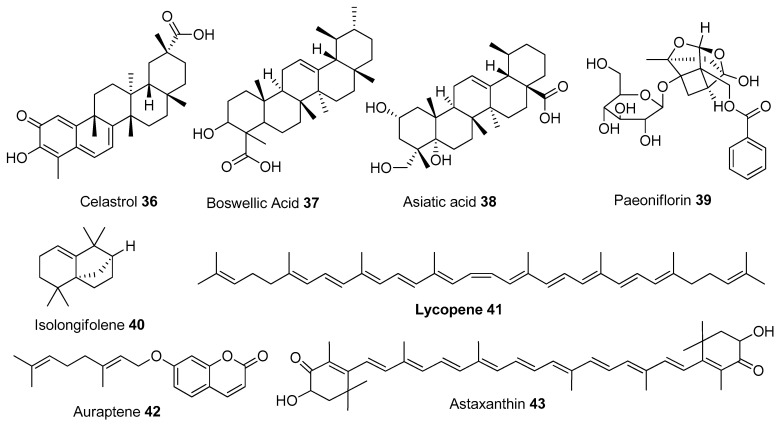
Chemical structures of terpenes.

**Figure 7 biomolecules-13-00226-f007:**
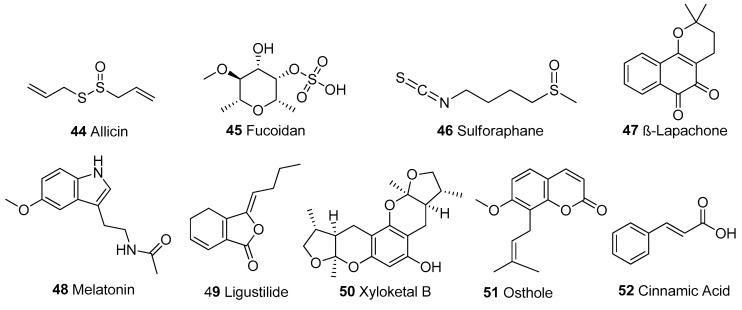
Other structural classes. Organosulphur compounds: 44-46; benzochromone: 47; Amine: 48; Lactone: 49; Cyclic polyketide: 50; Coumarin derivative: 51, Organic acid: 52.

**Figure 8 biomolecules-13-00226-f008:**
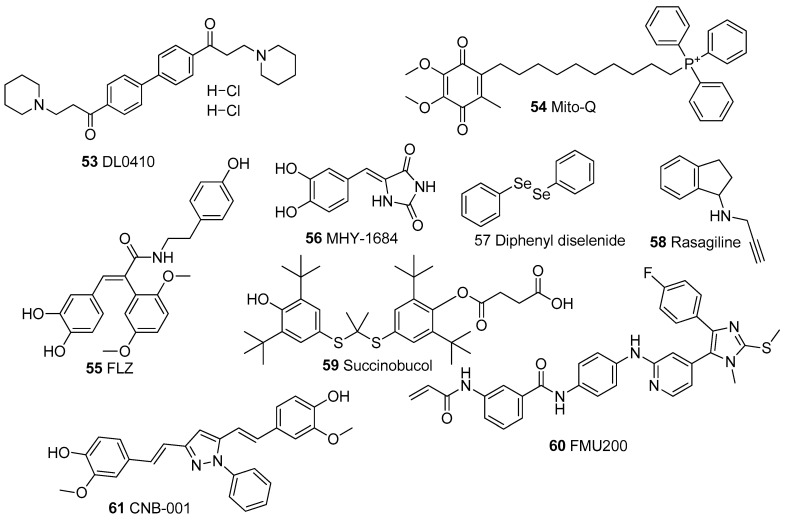
Chemical Structures of synthetic mitochondrial modulators.

**Figure 9 biomolecules-13-00226-f009:**
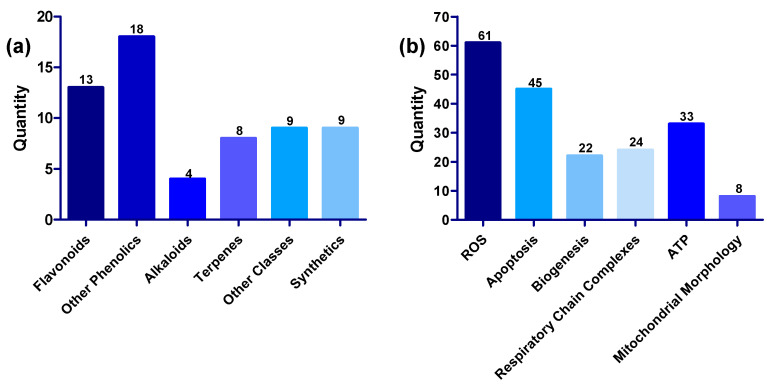
(**a**) Distribution of compounds across various structural classes of flavonoids, phenolic compounds, alkaloids, terpenes, other structural classes, and synthetic compounds. (**b**) Distribution of compounds across various targets, including ROS, apoptosis, biogenesis, respiratory chain complexes, ATP, and mitochondrial morphology.

**Table 1 biomolecules-13-00226-t001:** Mitochondrial Modulators.

	Compounds	Model	Dose	Mechanisms	References
1.	Quercetin	PC12 cells Wistar rats	1–100 μM 2–100 mg/kg	↓ROS, ↓LPO, ↑SOD, ↑GSH, ↑CAT, ↓Apoptosis, ↑MMP, ↑ATP, ↑Nrf2 ↑AMPK, ↑PGC-1α, ↑SIRT1, Restored Mt morphology, ↑PINK1 and ↑Parkin	[46,47,48,49]
2.	Baicalein	SH-SY5Y cells V79-4 cells	10–25 μM 10 μg/mL	↓ROS, ↓LPO, ↑SOD, ↓Bax, ↑Bcl-2, ↓Cyt c, ↓Caspase-3 ↓Ca^2+,^ ↑Nrf2, ↑MMP	[50,51,52]
3.	Kaempferol	HUVECs cells L2 cells	10–40 μM	↓ROS, ↓LPO, ↑SOD, ↑GSH, ↑GPx, ↓Bax, ↑Bcl-2, ↓Cyt c, ↓Caspase-3 ↑MMP, mPTP blockage, ↑ATP, ↑SIRT1	[53]
4.	Fisetin	Wistar rats	15 and 20 mg/kg	↓ROS, ↓LPO, ↑SOD, ↑GSH, ↑CAT, ↑MMP, ↓Caspase-3, ↑ATP, ↑Complex I	[54,55,56]
5.	Luteolin	C57BL/6 mice	10 μg/kg	↓ROS, ↓LPO, ↑MMP, ↓Caspase-3, -9, ↑ATP, ↑AMPK, ↑Complexes I, II, III, IV, and V	[57]
6.	7,8-dihydroxyflavone	Wistar Rat H92c cells	5–20 mg/kg 100 μM	↓ROS, ↓LPO, ↑SOD, GSH, CAT, GPx ↑MMP, ↑Complexes I, II, III, IV	[58,59]
7.	Naringenin	SH-SY5Y cells	10–80 μM	↓ROS, ↓LPO, ↑SOD, ↑GSH, ↑CAT, ↑PI3K/Akt/GSK-3β, ↑MMP, ↑ATP, ↑Nrf2, ↑Complexes I, V	[60,61,62,63]
8.	Genistein	C57/BL6J mice H9c2 cells	2.5–10 mg/kg 10 pM–1 μM	↓ROS, ↓LPO, ↑SOD, ↑GSH, ↑CAT ↑GPx ↑MMP, ↓Cyt c, ↓Caspase-3, ↑Nrf2	[64,65,66]
9.	Icariin	Human NP cells	10 μM	↓ROS, ↓Bax, ↑Bcl-2, ↓Cyt c, ↓Caspase-3, ↑MMP, ↑Nrf2	[67]
10.	Catechin	EA.hy926 cells HepG2 cells	4 mM 10 μM	↓ROS, ↓LPO, ↑SOD, ↑CAT, ↑MMP, Restored Mt morphology, ↑SIRT1, ↑Complex I, ↑ATP	[68,69,70]
11.	Epicatechin	MRC-5 cells BV2 cells	10 μM 100 μM	↓ROS, ↓LPO, ↑SOD, ↑CAT, ↑MMP Restored Mt morphology, ↑AMPK, ↑SIRT1, ↑Complex I, ↑ATP	[68,71]
12.	Epigallocatechin gallate	HLE B-3 cells	50 μM	↓ROS, ↓LPO, ↑SOD, ↑GSH, ↑CAT ↑GPx, ↓Bax, ↑Bcl-2, ↓Cyt c, ↓Caspase-3, -9, ↑MMP, ↑ATP	[72,73]
13.	Hesperidin	Mice	25–50 mg/kg	↑SOD, GSH, CAT, GPx, ↓Caspase-3, -9, ↑MMP, ↑ATP, ↑Complexes I, II, IV, V	[74,75]
14.	Resveratrol	Wistar rats C57BL/6 mice MC3T3-E1 cells	20 mg/kg 40 mg/kg 25 μM	↓ROS, ↑SOD,↑Bcl-2, ↓Cyt c, ↑SIRT1-AMPK-PGC-1α, ↑PINK1 ↑MMP, ↑Nrf2, ↑ATP, ↑Complex I, CypD	[76,77,78]
15.	Tyrosol	CATH.a cells SH-SY5Y cells	50–200 μM	↓ROS, ↑MMP ↓Bax, ↑Bcl-2, ↓Cyt c, ↓Caspase-3, -9, ↑ATP	[79]
16.	Hydroxytyrosol	ARPE cells HCN-2 cells	100 μM 30 μM	↓ROS ↓LPO, ↑SOD, GSH, CAT, GPx ↓Ca^2+,^ ↑Nrf2, ↑MMP, ↑Complexes I, II, V	[80,81]
17.	Curcumin	SH-SY5Y cells	5 μM	↓ROS, ↓LPO, ↑GSH, ↑GPx, ↑MMP, ↑ATP, ↓Ca^2+,^ ↓Caspase-3, -9, ↑Complexes II, IV	[82,83]
18.	Esculetin Mito-esculetin	C2C12 cells HAEC cells	5 μM 2.5 μM	↓ROS, ↑GSH, ↑MMP, ↑Nrf2 ↓Caspase-3, -8, ↑AMPK/SIRT3/ PGC1α	[84]
19.	Protocatechuic Acid	PC12 cells Human Platelets	0.1–1.2 mM	↓ROS, ↓LPO, ↑GSH, ↑GPx, ↑MMP ↓Bax, ↑Bcl-2, ↓Cyt c, ↓Caspase-3, -9, ↓PI3K/Akt/ GSK-3β	[85,86,87]
20.	Salvianic acid A	SH-SY5Y cells SD Rat	1–100 μg/mL 52 μg/mL	↓ROS, ↓LPO, ↑MMP ↓Bax, ↑Bcl-2, ↓Cyt c, ↓Caspase-3	[88,89]
21.	Sesamin	BEAS-2B cells	40 μM	↓ROS, ↓LPO, ↑SOD, ↑CAT, ↓Bax, ↑Bcl-2, ↓Caspase-3 ↑MMP, ↑Nrf2, ↓PINK1, Parkin	[90]
22.	α-Arbutin	SH-SY5Y cells	1–100 μM	↓ROS, ↑SOD, ↑GSH, ↑MMP, ↑ATP, ↓AMPK	[91]
23.	Ellagic acid	Wistar rats C57BL/6 mice SH-SY5Y cells	10–100 mg/kg 20 μM	↓ROS, ↑SOD↑, MMP, ↑Nrf2 mPTP blockage, ↓Cyt c, ↑ATP, ↑Sirt3 ↑Complexes I, II, III, and IV	[92,93,94,95,96]
24.	Mangiferin	SH-SY5Y cells C57BL/6 mice	10–50 μM 10–50 mg/kg	↓ROS, ↓LPO, ↑SOD, ↑GSH ↑CAT, ↑GPx, ↓Bax, ↑Bcl-2, ↓Cyt c, ↓Caspase-3, -9, ↑Nrf2, ↑MMP, ↑ATP, ↑Complex I	[97,98,99]
25.	Panduratin A	RPTEC/TERT1	5 μM	↓ROS, ↑MMP, ↑Bcl-2, ↓Cyt c, ↓Caspase-3	[100]
26.	α-Mangostin	SH-SY5Y cells	0.03–0.3 μM	↓ROS, ↑MMP, ↑ATP, ↓Caspase-3, -8	[101]
27.	Rosmarinic acid	H9c2 cells SH-SY5Y cells Zebra fish C57BL/6 Mice	1–200 μM 20–80 mg/kg	↓ROS ↑GSH, ↑Nrf2, ↑MMP, ↑SIRT1/PGC-1a ↑PI3K/Akt Restored Mt Morphology, ↑ATP	[102,103,104,105]
28.	Chlorogenic acid	HUVECs cell Albino mice	25–160 μM 50 mg/kg	↑SOD, ↑GSH, ↑MMP, ↑ATP, ↑SIRT1 ↓Caspase-3, ↑Complexes I, II, III, IV, and V	[106,107,108]
29.	Theaflavin	TCMK-1 cells	2–10 μM	↓ROS, ↓LPO, ↑SOD, ↑MMP, ↑Nrf2, ↓Bax, ↑Bcl-2, ↓Caspase-3 ↑ATP, Restored Mt morphology	[109]
30.	Salvianolic acid A	Cardiomyocyte 3T3-L1 cells	12.5–50 μg/mL 1–100 nM	↓ROS, ↓Bax/Bcl-2 ratio, ↓Caspase-3, ↑Akt/GSK-3β, ↑MMP mPTP blockage, ↑ATP, ↑PGC-1α, ↑Complexes III and IV	[110,111]
31.	Salvianolic acid B	HL-7702 cells IEC-6 cells	50–200 μM 2.5–40 μM	↓ROS, ↓LPO, ↑SOD, ↑CAT, ↑MMP, ↑PI3K/Akt/GSK-3β, ↑ATP Restored Mt morphology, ↑AMPK/Sirt3	[112,113]
32.	Sarain A	SH-SY5Y cells	0.01–10 μM	↓ROS, ↑SOD, ↑Nrf2, ↑MMP, mPTP blockage, ↓Cyp D, ↑ATP	[114]
33.	Sarain 2	SH-SY5Y cells	10 μM	↓ROS, ↑MMP	[114]
34.	Piperine	Wistar rats	10 mg/kg	↓ROS, ↓LPO, ↑GSH, ↑MMP, ↑Bax/Bcl-2, ↓Cyt c, ↓Caspase-3, -9 ↑Complexes I, II, and II	[115,116,117]
35.	Caffeine	SH-SY5Y cells SD rats APPsw rats	1–100 μM 40 mg/kg 120 mg/L	↓ROS, ↑MMP, ↑ATP ↓Bax/Bcl-2, ↓Cyt c, ↓Caspase-3, -6 ↑PI3K/Akt, ↑Complexes I, II, and III	[118,119,120,121]
36.	Celastrol	C57BL6/J mice SD rats	100 μg/kg 1–3 mg/kg	↓ROS, ↑GSH, ↑MMP, ↑ATP ↑AMPK/SIRT1/ PGC1α, ↑Complexes I, and III	[122,123]
37.	Boswellic acid	Albino rats	100–250 mg/kg	↓ROS, ↓LPO, ↑SOD, ↑GPx, ↑CAT, ↑Nrf2, ↓Caspase-3 ↑Complexes I, II, III, and IV	[24,124]
38.	Asiatic acid	SH-SY5Y cells	10 nM	↓ROS↑, MMP, ↓Bax, ↑Bcl-2, ↓Cyt c, ↓Caspase-3, -8, -8, -9	[125]
39.	Paeoniflorin	SD rats, PC12 cells	25–100 μM	↓ROS, ↑MMP, ↑ATP, ↓Bax, ↑Bcl-2, ↓Cyt c, ↓Caspase-3, -9	[126,127,128]
40.	Isolongifolene	Rats SH-SY5Y cells	10 mg/kg 10 μM	↓ROS, ↓LPO, ↑SOD, GSH, CAT, GPx, ↑Bax, Bcl-2, ↓Cyt c, ↓Caspase-3, -6, -8, -9, ↑PI3K/Akt/ GSK-3β, ↑MMP	[129,130]
41.	Lycopene	SD rats	5 μM	↓ROS, ↑MMP, mPTP blockage, ↑ATP, ↓Bax, ↑Bcl-2, ↓Bax/Bcl-2 ratio, ↓Cyt c, ↓Caspase-3, -9, ↑PGC1α, ↑Complexes I, II, III, IV	[131]
42.	Auraptene	SN4741 cells bEnd.3 cells	10 μM 1 μM	↓ROS, ↑MMP, ↑Nrf2	[132,133]
43.	Astaxanthin	LO2 cells	30–90 μM	↓ROS, ↑MMP, ↑ATP, ↓Bax, ↓Caspase-3, ↑PGC1α, Restored Mt morphology	[134]
44.	Allicin	PC12 cells	0.01–1 μg/mL	↓ROS, ↑MMP, ↑PGC1α, ↓Bax, ↑Bcl-2, ↓Cyt c, ↓Caspase-3	[135,136]
45.	Fucoidan	SH-SY5Y cells HPBM cells	50 μg/mL 20 and 50 μM	↓ROS, ↑MMP, ↓Bax, ↑Bcl-2, ↓Caspase-3 ↑Complexes I and IV, ↑AMPK/PGC1α	[137,138]
46.	Sulforaphane	HHL-5 cells	10 and 250 μM	↓ROS, ↓LPO, ↑SOD, ↑GSH, ↑Nrf2, ↑MMP, ↑ATP, ↓Apoptosis, ↑PGC1α, ↓Ca^2+,^ ↑Complexes I and IV	[139,140]
47.	β-Lapachone	MELAS cells	1 μM	↓ROS, ↑MMP, ↑ATP	[141]
48.	Melatonin	Porcine oocytes	500 nM	↓ROS, ↑MMP, ↑ATP,↓Caspase-3 ↑PGC1α/SIRT1	[142]
49.	Ligustilide	HT-22 cells, SD rats	20 μM, 10, 20 mg/kg	↓ROS, ↑MMP, ↑PINK1/Parkin	[143]
50.	Xyloketal B	PC12 cells	100–250 μM	↓ROS, ↑GSH, and ↑MMP Restored Mt morphology	[144,145]
51.	Osthole	PC12 cells SD rats	7 μg/cc 50 mg/kg	↓ROS, ↑MMP, ↑ATP, ↓Bax, ↑Bcl-2, ↓Bax/Bcl-2 ratio, ↓Cyt c, ↓Caspase-3, -9, ↑Complexes I, II, III and IV	[146,147]
52.	Cinnamic Acid	H9c2 cells	100–500 nM	↓ROS, ↓LPO, ↑SOD, ↑GSH, ↑MMP, ↓Bax, ↑Bcl-2, ↓Caspase-3	[148]
53.	DL0410	SH-SY5Y cells	1–10 μM	↓ROS, ↓LPO, ↑Nrf2, ↑MMP, ↓Bax, ↑Bcl-2, ↓Cyt c, ↓Caspase-3	[149]
54.	Mito-Q	NP cells	500 nM	↓ROS, ↓LPO, ↑SOD, ↑GSH, ↑Nrf2, ↑MMP, ↓PINK1/Parkin	[150]
55.	FLZ	SH-SY5Y cell	100 μM	↓ROS, ↑GSH, ↑MMP, mPTP blockage, ↑Complexes IV	[151,152]
56.	MHY-1684	hCPCs^c-kit+^	1 μM	↓ROS, ↑AKT signaling, ↓Apoptosis	[153]
57.	Diphenyl diselenide	HT22 cells LDLr^−/−^ mice	2 μM 1 mg/kg	↓ROS, ↓LPO, ↑SOD, ↑GSH, ↑GPx, ↑MMP ↑Complexes I and II	[154,155,156,157]
58.	Rasagiline	SH-SY5Y cells Rat mitochondria	100 nM 1–10 μM	↓ROS, ↑SOD, ↑GSH, ↑MMP, ↓Cyt c, ↑ATP	[158,159,160,161]
59.	Succinnobucol	SH-SY5Y cells	3 μM	↓ROS, ↑GSH, ↑MMP, ↓Cyt c, ↑ATP	[162]
60.	FMU200	SH-SY5Y cells	0.1 and 1 μM	↓ROS, ↑MMP	[163]
61.	CNB-001	C57BL/6 mice SK-N-SH cells	6–48 mg/kg 2 μM	↓ROS, ↓LPO, ↑SOD, ↑GSH, ↑GPx,↑CAT, ↑MMP ↓Bax, ↑Bcl-2, ↓Cyt c, ↓Caspase-3	[164,165]

## Data Availability

Not applicable.

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
