# Peer review of "Mitochondrial Modulators: The Defender"

_biomolecules, 2023, doi:10.3390/biom13020226_

Round 1
Reviewer 1 Report
In this manuscript, Makinde et al. highlight the main functions of mitochondria in both health and disease conditions, also showing explanatory pictures. Moreover, the authors provide a comprehensive insight into compounds with therapeutic potential on mitochondrial functions and their mechanisms of action, with a focus on compounds that can modulate the mitochondria such that mitochondrial dysfunction is mitigated or prevented altogether, conducting an extensive literature search. Although the work is clear and well written, and the bibliography appropriate, I believe that small revisions should be made.
Minor revisions:
Line 47: It is highly recommended the addition of the following reference “ Mitochondria and Sex-Specific Cardiac Function.” doi: 10.1007/978-3-319-77932-4_16. PMID: 30051389”.
Line 115: it is highly recommended the addition of the following reference at the end of the sentence, “Biomarkers of Oxidative Stress in Metabolic Syndrome and Associated Diseases. doi: 10.1155/2019/8267234. PMID: 31191805.”
Overall, there are several typos throughout the text
Author Response
Dear reviewer,
Thank you very much for your valuable comments to the manuscript, our response is attached here.
Regards,
Yun

Reviewer 2 Report
Mitochondria are dynamic organelles that undergo cycles of homotypic fusion and fission events, which are believed to play an important role in controlling organelle number, subcellular distribution, morphology, and ATP production. Recent studies in model animals demonstrated that genetic ablation of individual components involved in mitochondrial dynamics impairs organ function and its whole body metabolism. Abnormal mitochondrial dynamics are also related to several human diseases such as Charcot-Marie-Tooth disease type 2A (CMT2A) and dominant optic atrophy (DOA). Although the importance of mitochondrial dynamics is well documented, the molecular mechanism and its pathogenesis remain unclear. In the manuscript, Makinde et al have reviewed importance of physiologic and pathologic functions of mitochondria and discussed roles of bioactive compounds with its mechanistic actions against the organelle. I think the topics they mentioned in the text are potentially interesting and the logic of the manuscript is also straightforward. I recommend the manuscript is suitable for publication in the journal, and a couple of major & minor issues/comments will further strength their conclusion.
Major issue
Mitochondria are also known as signaling platform for innate immunity against infectious microorganisms. The authors should mention more about how some chemical compounds involved in the immune function via mitochondrial pathway (See following comments).
Minor issues
1). Cyclosporin A is well known compound involved in MTP inhibitor.
2). Fluorizoline is found to be a prohibitin (PHB)-binding compound (PMID: 34580273).
3). Aurilide is a marine natural compound that binds to PHB (PMID: 21276946).
4). Prohibitin (PHB) is a mitochondrial inner membrane protein that links to a myriad of mitochondrial function. A recent work has investigated their structural-functional role of PHB involved in innate immunity (PMID: 31522117). I guess discussing the functional role of PHB involving in mitochondria-mediated in innate immunity would be nice in the revised MS.
Author Response
Dear Reviewer,
Thank you very much for your valuable comments to the manuscript. Our responses are attached here.
Regards,
Yun

Round 2
Reviewer 2 Report
N/A